# Synthesis of Azacalixarenes and Development of Their Properties

**DOI:** 10.3390/molecules26164885

**Published:** 2021-08-12

**Authors:** Hiroyuki Takemura

**Affiliations:** Department of Chemical and Biological Sciences, Faculty of Science, Japan Women’s University, Mejirodai 2-8-1, Bunkyo-ku, Tokyo 112-8681, Japan; takemurah@fc.jwu.ac.jp

**Keywords:** azacalixarene, calixarene, macrocycle, cyclophane, supramolecule

## Abstract

This review focuses on the synthesis, structure, and interactions of metal ions, the detection of some weak interactions using the structure, and the construction of supramolecules of azacalixarenes that have been reported to date. Azacalixarenes are characterized by the presence of shallow or deep cavities, the simultaneous presence of a basic nitrogen atom and an acidic phenolic hydroxyl group, and the ability to introduce various side chains into the cyclic skeleton. These molecules can be given many functions by substituting groups on the benzene ring, modifying phenolic hydroxyl groups, and converting side chains. The author discusses the evidence of azacalixarene utilizing these characteristics.

## 1. Introduction

### 1.1. General Concepts of the Azacalixarenes and Related Compounds

Calix[n]arenes are cyclic compounds consisting of alternating *p*-substituted phenol and methylene chains. As they are easy to synthesize and obtain in sufficient quantities for laboratory use, their structures are unique, and various applications are possible by modifying the molecules; research on them was, at one time, widely conducted worldwide, and all kinds of research, from basic to applied, was carried out [1,2]. As a result, compounds with similar structures have appeared one after another, and a number of cyclic compounds bearing the name calixarene have been reported to date. However, some of them should be classified as [1^n^]metacyclophanes, and in order to be given the name “calix”, the aromatic rings of the molecule must have a cup or dish-like structure with concave cavities. In addition, the study of cyclic compounds, such as resorcinarene and pyrogallarene, also became popular. In particular, cryptophanes and carcerands have developed interesting chemistry based on resorcinarene [3].

Molecules in which several -CH_2_N- units are inserted into the methylene of calixarene are called homoazacalixarenes, and various homologs are formed depending on the number and position of the inserted -CH_2_N- units.

Similarly, thiacalixarenes, in which the methylene of the calixarene is replaced by a sulfur atom [4,5,6,7,8]; azacalixarenes, in which the methylene is replaced by a nitrogen atom [9,10,11,12,13,14,15,16,17,18,19]; and homoxacalixarenes, in which a -CH_2_O- unit is inserted [20,21], have been widely studied. However, in this review, only homoazacalixarenes are considered, referred to as azacalixarene, not homoazacalixarene, unless it is necessary.

### 1.2. The Birth of Azacalixarene

The author worked on azacyclophane and host–guest chemistry in the 1980s when calixarene chemistry was in its infancy. The author found structural formula **1** in Figure 1 in the literature of Burke et al. [22]. This article describes the synthesis of benzoxazine derivative **2** by the reaction of various amines with phenol and formalin. Additionally, in one of the attempts, it was mentioned that **1** could be formed by the reaction of 2,6-bis(hydroxymethyl)cresol with benzylamine. However, in the reaction of reflux benzene (3 h), 86% of 2,6-bis(hydroxymethyl)cresol was recovered, and compound **1** was not detected.

This structure, [3]metacyclophane, seems to be clearly distorted to cyclophane researchers. Therefore, it seems obvious that it cannot be produced by their synthetic method. In fact, Burke et al. experimentally ruled out the structure of **1**. Furthermore, it is clear to the author that the OH signal is not as broad as it should be, even for oligomers. Moreover, if it were acyclic, a terminal signal would be shown, but this was not found. The symmetry of the spectra also confirmed that the structure was cyclic. Then, the author estimated that if it were to form, it would be a trimer. The structure of **1** was denied but very attractive. In an attempt, *p*-methyl-bis(hydroxymethyl)phenol was refluxed with benzylamine in toluene and heated for two days while removing the resulting water with a Dean–Stark condenser. The resulting reaction mixture was a highly viscous gum-like substance. The reaction product was mixed with equal amounts of acetone and methanol and stirred vigorously for a while to obtain a pale yellow powder. NMR showed that it was a cyclic material; that is, the signal of the phenolic hydroxyl group was very broad and appeared in an unusually low field (11.2 ppm), supporting intramolecular hydrogen bonding in a cyclic structure. FABMS showed trimeric molecular ion peaks (and a few tetramers), and elemental analysis confirmed the composition [23]. Subsequently, amino group-modified silica gel was found to be very good for the purification of homoazacalix[n]arene. Ebata et al. confirmed that phenol forms a cyclic trimeric structure in the gas phase by IR-UV double-resonance and stimulated Raman–UV double-resonance spectroscopy [24]. Therefore, it is natural that the azacalix[3]arene structure is formed in non-polar solvents.

### 1.3. Nomenclature

In the case of azacalixarenes, a simplified nomenclature is necessary, as there are many possible structures considering the amount of nitrogen, the number of phenolic units, and the position of the nitrogen atom. For example, in the case of compounds **A** and **B** (Figure 2), both are tetrahomodiazacalix[6]arene, but the structures are different. Therefore, we use a nomenclature that is intuitive and easy to understand, borrowing some of the nomenclatures of cyclophanes.

In this regard, assume a coordinate axis with the origin at the center of the compound, place the heteroatom in the first quadrant, and indicate the number of bridging atoms in a clockwise direction in brackets. If there are multiple nitrogen atoms, the group of atoms containing as many nitrogen atoms as possible should be placed at the beginning of the brackets. In other words, in the case of compound **C**, the nitrogen atom is placed in the upper right corner, and the number of bridging atoms connecting the phenol units is placed in brackets, resulting in [3.1.1.1]. In this case, one can call it azacalix[3.1.1.1]arene. So compounds **A** and **B** become diazacalix[3.3.1.1.1]arene and diazacalix[3.1.1.3.1.1]arene, respectively.

## 2. Synthesis and Structures

### 2.1. Procedures

The skeletons of cyclic compounds **6**, **7**, and **8** were readily obtained by condensation of bis(hydroxymethyl)phenols **3**, **4**, and **5** with benzylamines (Scheme 1) [25]. By changing the benzylamines used, the side chains can be functionalized for other purposes. Initially, non-polar solvents, such as toluene and xylene, were accidentally used in this reaction, but this resulted in the formation of cyclic products. When a polar solvent such as pyridine was added to toluene, only polymer-like products were formed. This is due to the self-assembly of cyclic compounds in non-polar solvents by hydrogen bonding between hydroxyl groups or with amines [23,25].

The reaction time was 72 h, but in most cases, the reaction was almost over in approximately 24 h. A variety of benzene derivatives and heterocycles as side chains could be introduced, and the yields were satisfactory (Table 1) [26].

As another synthetic method, one can construct the calixarene structure from the basic starting materials. A mixture of aqueous methylamine, formalin, phenol, and potassium hydroxide was heated at 50–55 °C for 48 h, and the viscous resinous material thus obtained was then removed. The resinous material was then refluxed in xylene for 48 h, and water was removed, thus resulting in the *N*-Me derivatives **8**, **9**, and **10** (Scheme 2).

In addition to this synthetic method, that of azacyclophanes [27] was also applied to obtain *N*,*N*′-ditosyl-*p-tert*-butyl-tetrahomodiazacalix[6]arene **12** by methylation of the phenol trimer followed by bromomethylation and cyclization with *p*-TsNH_2_ as a nitrogen source (Scheme 3). This was the case when the conventional azacyclophane method was applied, but this method was time-consuming, the process of deprotection of *N*-Ts and OMe groups was problematic, and it was found to be unsuitable as a synthetic method for azacalixarenes. Compound **12** is a byproduct of the attempted synthesis of the smallest number of azacalixarenes, *N*-tosyl-*p-tert*-butyl-azacalix[3.1.1]arene **11**. The synthesis of compound **11** has not yet been achieved. Here, the phenol substituent in some compounds is a *tert*-butyl group, but it is not necessary to use this group. The *para*-position of phenol can be methyl, chloro, or bromo as azacalixarenes have high solubility. The introduction of the *tert*-butyl group makes the azacalixarens too soluble and may cause problems. However, the high solubility of azacalixarenes is not due to the molecule being more flexible than that of calix[n]arene but due to the presence of side chains. On the contrary, the cone conformation of azacalix[3.1.1.1]arene due to intramolecular hydrogen bonds was found to be more rigid than calix[n]arene (see below). In addition, as discussed later, those with a large ring and an *N*-Me group have poor solubility.

### 2.2. Synthesis and Modification of N-Methyl-Diazacalix[3.1.3.1]arene

The hydroxyl group of calixarenes and the *para*-substituents of phenols have been modified in many ways. Azacalixarenes are structurally characterized by the presence of a nitrogen atom, which allows the introduction of various side chains to the nitrogen atom. Furthermore, nitrogen can be quaternized, which can add a positive charge. Phenolic OH has a proton that dissociates easily, so depending on the pH, it is possible to have both positive and negative charges in the molecule at the same time. The following molecule was synthesized in the hope that it would be able to recognize substances such as amino acids, which can also have positive and negative charges [28]. The reason why the nitrogen side chain is a methyl group is due to the fact that benzyl group alkylation is difficult, and the NMR spectrum tends to be complicated. The synthesis of **7** was accomplished by mixing methylamine solution, alkali, and phenol derivatives and heating. The target product, *N*,*N*′-dimethyl-diazacalix[3.1.3.1]arene **7** (X = Me, R = Me), was obtained in 44% yield when 10 times as much methylamine was used as the phenol derivative. When the amount of methylamine was lower than five times, **7** (X = Me, R = Me) and azaoxacalix[3.1.3.1]arene **13** were obtained in 28.6 and 23.0% yields, respectively. Subsequently, the hydroxyl group of **7** was protected with an acetyl group, followed by *N*-alkylation and deprotection to afford the desired product. This compound produced azacalix betaine **14** upon treatment with a weak base, such as sodium bicarbonate (Scheme 4). Unfortunately, it did not form complexes with a number of amino acids.

### 2.3. Reaction between Bis(hydroxymethyl)phenols and Diamines

Sone et al. reported an amino acid cross-linked azacalixarene (Figure 3) [29,30,31]. The synthesis was based on the direct reaction of bis(chloromethyl)phenol oligomers with amino acids. The molecular recognition of chiral amines was also reported [32].

Similar cross-linked azacalixarenes were also synthesized. When diamines such as *p*-xylylene diamine or *m*-xylylene diamine were used instead of benzylamine and reacted with bis(hydroxymethyl)phenol **4** (X = Me), the nitrogen-bridged compounds **15** and **16** were obtained (Scheme 5). Here, the reaction of *m*-Xylylenediamine with a phenol derivative **18** consisting of three phenolic units produced a low yield of cyclic products, and more compound **16** with one less phenolic unit were obtained than the expected compound **17** [33]. Thus, in these reactions, the phenol unit is sometimes cleaved, but it was found that cleavage of the phenol unit often occurs, as in the reaction described below.

Reactions of phenol dimers or trimers with various diamines yielded *N*,*N*′-bridged azacalixarenes **19**–**21**, as shown in Scheme 6. The structure of compound **20** was confirmed by X-ray crystallography [34]. Here, when the bis(hydroxymethyl)phenol derivative has four phenolic units, the products can be bridged type **A** and linked type **B** in a reaction with diamine (Scheme 7).

As shown in Scheme 8, when phenol tetramer **5** (X = Me) was used as a phenol derivative and reacted with various diamines, the product was not found to be the cross-linked type **A**, but structure **B**, in which two azacalix[3.1.1.1]arenes are linked [34].

When phenol tetramer **5** was reacted with *p*-Xylylenediamine and 1,6-diaminohexane, the isolated reaction product was a single component. Compound **22** was recrystallized from CH_2_Cl_2_/CH_3_CN and obtained as yellow crystals in 60% yield. Compound **23** was recrystallized from CH_2_Cl_2_/CH_3_CN and obtained as yellow crystals in 51% yield. The reaction with diamine (1,2-bis(2-aminoethoxy)ethane) produced only compound **24** in 19% yield. The structure of compound **22** was determined by crystallographic analysis (Figure 4), and the structures of **23** and **24** were determined by NMR and theoretical calculations (Scheme 8). The reaction of tris(3-aminoethyl)amine with bis(hydroxymethyl)phenol dimer afforded the disk-shaped compound **25a** in 10% yield. The structure was clarified by crystallographic analysis. The reaction of tris(3-aminopropyl)amine with bis(hydroxymethyl)phenol dimer afforded compound **25b** in a 14.4% yield. The crystallization of this compound was attempted from various solvents, but only powder was obtained. In the molecular model, the central nitrogen lone pair is less distorted when it is facing outward.

### 2.4. Synthesis of New Azacalixarenes via Dihydro-1,3-benzoxazines Derivatives [35]

In 1965, Burke and co-workers showed that dihydro-1,3-benzoxazines react with phenols at room temperature to produce *N*,*N*-bis(2-hydroxy-l-phenylmethy1)methylamine. They stated that this reaction could be used as a new aminoalkylation of phenol [36].

As an example, the case of *o*-cresol is shown in Scheme 9. In the original report, the reaction was carried out in methanol at room temperature for several months, although the yields were satisfactory in many cases. If this reaction was carried out at 80 °C for 6 days, the yield was low. However, this reaction can be applied to the synthesis of azacalixarenes with structures that could not be synthesized in a single step until now. For example, many benzylamine-based side chains have been used as synthetic units, but NMR spectra show that the methylene of the cyclic structure overlaps with the methylene of the benzyl side chain, causing an inconvenience. The advantage of this method is that it eliminates this inconvenience and allows the introduction of a wide variety of side chains, such as volatile amines, e.g., methylamine, which could not be used in previous synthetic methods. To date, only a limited number of *N*-Me products have been obtained by the one-step synthesis method, namely, *p*-methyl-*N*,*N*′-dimethyl-diazacalix[3.1.3.1]arene **7**, *p-tert*-butyl-*N*-methyl-azacalix[3.1.1.1] arene **8**, *p-tert*-butyl-*N*,*N*′,*N*”-trimethyl-triazacalix[3.3.3.1]arene **9**, and *p*-*tert*-butyl-*N*-methyl-azacalix [3.1.1.1.1.1]arene **10** (Scheme 2). Furthermore, this stepwise cyclization method can be used to synthesize azacalixarenes that are asymmetric on the N∙∙∙N′ axis. Additionally, the reaction of bis(hydroxymethyl)phenol tetramer **5** with benzylamine yielded only azacalix[3.1.1.1]arene, but not its dimer, azacalix[3.1.1.3.1.1]arene (described below). However, using Burk’s method, this is possible.

For these reasons, this synthetic method was applied for several reactions. The dihydrobenzoxazine derivatives **26** were obtained as crystalline or glassy substances by heating the phenol dimer, trimer, and tetramers with excess formalin and 40% methylamine solution in dioxane at 80 °C overnight (Scheme 10). These substances could not be purified as they became sticky and resinous with only a small amount of solvent, but they were so pure that purification was not necessary. In silica gel chromatography, some of them decomposed.

Dihydrobenzoxazine derivative **26a** and 2.2′-methylenebis(4-methylphenol) were heated under reflux in toluene to produce the desired *N*,*N*′-dimethyl-*p*-methyl-diazacalix[3.1.3.1]arene **7** (X = Me, R = Me) in high yield (Scheme 11). However, when phenol trimer or tetramer and their benzoxazine derivatives were reacted, cleavage of the phenolic units occurred at the toluene reflux temperature, resulting in a mixture of azacalixarenes. Therefore, the reaction was also studied for benzene boiling temperature and 40 °C. The products at each temperature for several combinations of phenol and its dihydrobenzoxazine derivatives were examined, and the results are shown in Scheme 12, Scheme 13, Scheme 14 and Scheme 15. Especially at high temperatures, *N*-Me-azacalix[3.1.1.1]arene **8** was the main product in the reaction when the phenolic units were numerous. Azacalixarenes **7** and **8** seem to be thermodynamically stable and are easily formed at high temperatures. In particular, the hydrogen bonding of phenol hydroxyl groups in the molecule seems to be the cause of their stabilization. The optimal structures of the methyl benzoxazine dimer, trimer, and tetramer were confirmed by Spiess et al. using ab initio calculations, and it was found that the structures are cyclic due to hydrogen bonding inside the molecules [37]. The results of their calculations confirm our experimental observations. The cleavage of phenolic units seems to occur relatively easily, and this was confirmed in the previous synthesis of *N*,*N*′-xylylene-bridged azacalixarenes (described above) [33]. Yamato et al. also reported that such cleavage occurs during the synthesis of octahydroxy[2.1.1.1. 2.1.1.1]metacyclophane [38]. In our case, the examination of the structure of the product suggests that the -N(Me)-CH_2_O- and phenol-CH_2_-phenol units of the benzoxazine derivative are dissociated. For the basic mechanism of cleavage, please refer to the original paper [35]. Lowering the reaction temperature suppressed the cleavage, but the reaction time became extremely long. One week was required at benzene boiling temperature, and one month at 40 °C, but the reaction was not complete. Except for the cyclic compounds that were able to isolate, the other compounds were acyclic oligomers and linear polymers. However, it was found that the desired asymmetric azacalixarenes could be obtained in good yields if the optimum temperature and time were chosen.

## 3. Structure of Azacalixarenes

### 3.1. Intramolecular Hydrogen Bonding and Structure in Solution

The characteristic feature of azacalixarene is the signal of hydroxyl protons in the NMR spectrum. They appear as relatively broad signals at lower fields than the corresponding calix[n]arene, thiacalix[4]arene, and oxacalixarenes (Table 2) [1].

Very broad OH stretching vibrational bands were also observed in the IR spectra. These indicate that the intramolecular hydrogen bonds in azacalixarenes are stronger than those in the corresponding calixarenes. This is due to the participation of basic nitrogen atoms in the cyclic hydrogen bonds. The OH∙∙∙N is a stronger hydrogen bond than OH∙∙∙O. This hydrogen bond appears in different ways depending on the size and symmetry of the ring in NMR at low temperatures: in triazacalix[3.3.3], there was no change except for broadening, while in azacalix[3.1.1.1], the hydroxyl proton signal in NMR split at low temperatures, and several signals appeared at 17.2–9.5 ppm. This indicates that the symmetry of the hydrogen bond is broken and localized [40]. In *p*-xylylene bridged **15**, the OH signal was singlet at room temperature, but at lower temperatures, it reached coalescence temperature at −20 °C and split into two signals at 12.2 and 8.2 ppm at −60 °C. The signal at 12.2 ppm was attributed to the OH∙∙∙N, and the signal at 8.2 ppm to the OH∙∙∙O hydrogen bonds as the shift of the signal at 8.2 ppm is similar to that of tetrahomodioxacalix[4]arene and hexahomotrioxacalix[3]arene [1]. The strength of this hydrogen bond also affects the conformational change in the molecule. In sum, in azacalixarenes, the barrier of aromatic ring inversion (cone ↔ partial cone) is higher for azacalix[3.1.1.1]arene than for calix[4]arene (Table 2). When the conformational change in *N*-Me-azacalix[3.1.1.1]arene was followed by variable-temperature NMR, an inversion of the aromatic ring was observed, as seen in calix[4]arene, and it was found that the aromatic ring changed from the cone to cone conformation via 1,4-alternate, as shown in Figure 5 [41,42].

### 3.2. Crystal Structures of Azacalixarenes and Their Metal Complexes

Although the author was the first to report the synthesis of azacalixarenes **6**, the first *p*-methyl-*N*-benzyl-triazacalix[3.3.3]arene synthesized did not crystallize, and thus it took some time to clarify the structure of triazacalix[3.3.3]arene. Hampton et al. were the first to clarify the crystal structure of triazacalix[3.3.3]arene. They synthesized triazacalix[3.3.3]arene with -CH_2_COOCH_3_ side chains on the nitrogen atoms by a method that was different from the one we used [43]. The structure is a shallow-bottomed dish-like structure with attractive C_3_ symmetry. Subsequently, the author’s group clarified some of the structures, and in collaboration with Dr. Pierre Thuéry and co-workers, the structures of lanthanides and uranyl ion complexes of azacalixarenes were also clarified (described later) [44,45,46,47,48]. The structure of *p*-methyl-*N*-benzyl-triazacalix[3.3.3]arene could not be obtained, but the crystal structure of *p*-Cl-*N*-benzyl-triazacalix[3.3.3]arene could be confirmed (Figure 6) [47,48]. Similarly, the structures of *p*-Me-*N*-benzyl-diazacalix[3.1.3.1]arene, *p-tert*-butyl-*N*-benzyl-azacalix[3.1.1.1]arene, and *p*-methyl-*N*-benzyl-azacalix[3.1.1.1]arene were also confirmed. All of them are characterized by the cone type, and the protons of the hydroxyl group are hydrogen-bonded to the oxygen and nitrogen atoms in a fixed directional arrangement. In triazacalix[3.3.3]arene, one benzyl side chain blocks the cavity. In diazacalix[3.1.3.1]arene, the side chains face outward, and the intramolecular cavity is a shallow dish shape. In contrast, azacalix[3.1.1.1]arene has a deep cavity similar to that of calix[4]arene.

A DMF inclusion complex of *p-tert*-butyl-*N*-benzyl-azacalix[3.1.1.1]arene was obtained from the DMF solution. When *p*-methyl-*N*-benzyl-azacalix[3.1.1.1] was recrystallized from DMF in the same way, two molecules of DMF were encapsulated in two molecules of azacalixarene (Figure 7). In these complexes, CH∙∙∙π interactions were observed between DMF molecules and the aromatic ring of the azacalixarene ring skeleton (described below). It is interesting to note that the inclusion mode and the packing in the crystal were completely different depending on the side chain and the substituent at the *para*-position of the phenolic hydroxyl group. In the case of no solvent inclusion, π-π stacking between aromatic rings was observed in *N*-(4-picolyl)-azacalix[3.1.1.1]arene [49].

In metal complexes, lanthanide (Yb^3+^ and Nd^3+^) and uranyl ion complexes were obtained coordinated only to oxygen atoms of the ligands. Structures of the complexes, UO_2_^2+^[*p-tert*-butyl-*N*-methyl-azacalix[3.1.1.1]arene], UO_2_^2+^[*p*-Cl-*N*-benzyl-triazacalix [3.3.3]arene], UO_2_^2+^[*p*-Me-*N*-benzyl-diazacalix [3.1.3.1]arene], Yb^3+^[*p*-Me-*N*-benzyl-diazacalix [3.1.3.1]arene], Yb^3+^[*p*-Cl-*N*-benzyl-triazacalix [3.3.3]arene], and Nd^3+^[*p*-Cl-*N*-benzyl-triazacalix [3.3.3]arene], were clarified [44,45,46,47,48]. These complexes are characterized by the fact that complex formation occurs under neutral to basic conditions and that crystals can be prepared at the same time. A characteristic feature of these complexes is that the metal coordinates with the oxygen of the hydroxyl group, and the protons of the hydroxyl group are transferred to the nitrogen atoms. The UO_2_^2+^[azacalix[3.1.1.1]arene] has a symmetrical coordination structure, and UO_2_^2+^ coordinates with the hydroxyl group, the “bottom” of azacalix. The coordination axis also extends in the vertical direction. Since metal coordination provides a stronger cone-type fixation, molecular modification based on such a coordination form is also possible (Figure 8).

### 3.3. Complex Formation with Metal Ions

As the crystal structure analysis revealed that the metal coordination structure differs greatly depending on the ligand structure, one can speculate that it may be possible to selectively recover metal ions from a mixed system of many metal ions, such as seawater, by changing the ligand. We previously conducted extraction experiments of lanthanides using *N*-(2-picolyl)-triazacalix[3.3.3]arene. The extractability was high, but there was no relationship between ion size and extractability. However, a slight difference in the pH of the used buffer solution was found to be sensitive to extractability. This was also the case for the extraction of uranyl ions. The simultaneous presence of a nitrogen atom and a hydroxyl group in the molecule has a significant effect on the complexation due to the protonation–deprotonation equilibrium [26].

Since *N*-benzyl-triazacalix[3.3.3]arene **6** has a hydroxyl group and a nitrogen atom, it was expected to have an affinity for metal ions. The affinity of the metal ion was confirmed by classical extraction experiments. First, alkali and alkaline earth metal ions whose extraction rates depend only on ion size and charge were investigated. Figure 9a shows the extraction results (the vertical axis is the distribution ratio × 100: [A]_org_/[A]_aq._ × 100). The distribution coefficients of picrates in the chloroform–water system showed that the distribution coefficients of alkali and alkaline earth at pH = 6 were low (note that the vertical axis is the distribution coefficient × 100). Next, the extraction of uranyl ion was attempted. The extraction of uranyl ions and their recovery from seawater are well known from the experiments of Tabushi et al. with cyclic 1,3-diketone [50]. In the experiments, simple simulated seawater containing only uranyl ions and sodium chloride ([UO_2_^2+^] = 10 ppm = 3.7 × 10^−5^ moldm^−3^, [NaCl] = 0.50 moldm^−3^]) was prepared, and extraction was carried out at different pH. The results are shown in Figure 9b. The characteristic feature here is that azacalixarene, without any functional groups on the side chains, extracted uranyl ions with high selectivity at a pH of approximately 6~7. Moreover, the distribution ratio was also high.

Khan et al. introduced -(CH_2_)_n_-Se(Te)-Ar (n = 2 and 3) into the side chain of triazacalix[3]arene and investigated the affinity of metal ions such as Na^+^, K^+^, Mg^2+^, Ca^2+^, Ba^2+^, Co^2+^, Ni^2+^, Cu^2+^, Ag^+^, Zn^2+^, Cd^2+^, Pb^2+^, Fe^3+^, and UO_2_^2+^. By potentiometry, they found that the affinity was only for UO_2_^2+^ [51,52]. However, these side chains are not necessary as the UO_2_^2+^ ion is trapped by the hydroxyl groups of azacalixarenes, as proven by crystallographic analysis (described above). Although all kinds of heavy metal ions are mixed in actual seawater, it was shown that only uranyl and lanthanide ions could be extracted selectively in the presence of a large excess of other metal ions.

## 4. Synthesis of Linked Azacalixarenes Using Side Chains

### 4.1. Supramolecules via Covalent Bonds

The next step was to introduce a picolyl group into the side chain and alkylate it to create a system of two or three azacalixarenes linked together (Figure 10). *N*-(4-picolyl)-azacalix[3.1.1.1]arene was obtained in high yields of 76.3–85.4% from phenol tetramer **5** and 4-picolylamine in refluxing toluene for 24–72 h. Next, *N*-(4-picolyl)-azacalix[3.1.1.1]arene and 1,4-bis(bromomethyl)benzene or 1,3,5-tris(bromomethyl)benzene were heated in CHCl_3_/CH_3_CN = 1/1, *v*/*v*) under reflux for 24 h. The products were recrystallized from methanol to produce **33** and **34** in 62% and 57% yields, respectively. Compound **33** was recrystallized from methanol to produce granular crystals. The results of crystal structure analysis are shown in Figure 11.

In the crystal, the cross-linked xylylene moiety is sandwiched between two azacalixarene units, forming a compact self-inclusion structure. Although a capsule-type inclusion supramolecular system was assumed, the inclusion of neutral molecules such as DMF, C_60_, and pyrene could not be confirmed. In this case, it can be assumed that the self-inclusion type is stable and the capsule type is more unstable. No inclusion was observed in the triply-bridged compound **34**, which also seems to be a compact structure in which the mesitylene unit in the middle is self-incorporated into three aza[3.1.1.1]arene units [49].

### 4.2. Linkage of Azacalixarenes by Metal Coordination Bonds

Next, supramolecules using coordination bonds were also synthesized: *N*-(4-picolyl)-azacalix[3.1.1.1]arene and PdCl_2_(PhCN)_2_ were stirred overnight in THF, and the complex precipitated as creamy crystals. The product was a powder that was soluble to some extent in chloroform but insoluble in other solvents, such as DMF and MeOH [49]. According to the mass spectrum, dimer **35** was the main component, but trimer and tetramer, which replaced Cl, were also produced. It is interesting to note that both dimer and tetramer can be obtained by adding or subtracting the amount of the reagent to be reacted. Since the solubility is poor, it is necessary to change the substituent of azacalix to *tert*-butyl or replace Cl on Pd with triflate for crystal structure analysis (Scheme 16).

## 5. Inclusion of Neutral Molecules in Azacalixarenes

### 5.1. C-H∙∙∙π Interaction and Structure in DMF Inclusion Complexes

Crystallization of three azacalix[3.1.1.1]arenes with different substituents and side chains from a solution containing DMF yielded DMF-inclusion complexes with different inclusion modes. In these crystals, the DMF molecules in the azacalix cavities were stabilized by C-H∙∙∙π interaction, which is an old but increasingly important interaction in modern chemistry. In particular, in the field of biochemistry, C-H∙∙∙π interaction is among the most important interactions for the higher-order structure of proteins [53,54,55,56,57,58,59,60,61]. C-H∙∙∙π interactions are also a factor in determining higher-order structures in supramolecular chemistry and host–guest chemistry [62,63,64,65]. The universality of the CH···π interaction has recently been revealed by various modern methods, such as crystallography, NMR, and theoretical calculations [66,67,68,69,70,71,72,73,74,75]. Crystal structure database analysis shows that CH···π interaction is observed more frequently than OH···π and NH···π. Suezawa et al. stated that the aromatic rings with the shortest distance to CH are distributed at approximately 290 pm [56,61]. The same conclusion was reached by Umezawa et al. [76]. The crystallization of *p*-*tert*-butyl-*N*-benzyl-azacalix[3.1.1.1]arene from DMSO/DMF yielded crystals with 1:1 inclusion of DMF only [77]. The DMF inclusion compound was also obtained by crystallization from the CH_2_Cl_2_/DMF solution. As mentioned above, in *p*-methyl-*N*-benzyl-azacalix[3.1.1.1]arene, two molecules formed a capsule-type structure and encapsulated two molecules of DMF (Figure 7). Since the inclusion pattern was found to be different depending on the substituent, a *tert*-butyl group at the *para*-position of phenol and a methyl group on the nitrogen were introduced next. The synthesis of *p-tert*-butyl-*N*-methyl-azacalix[3.1.1.1]arene was carried out in one step, i.e., from *p-tert*-butylphenol, methylamine solution, and formalin in 29.3% yield [33]. The DMF inclusion compound was obtained as a yellow prism when crystallized from CH_2_Cl_2_/DMF solution. As expected, the packing behavior in the crystal was very different when the substituents were changed (Figure 12a). Another feature is the intramolecular pattern of hydrogen bonding between the phenolic hydroxyl group and the nitrogen atom. Only in the *N*-Me derivative, as seen in some examples of azacalixarenes, does the proton of one of the phenolic hydroxyl groups dissociate and protonate to the nitrogen atom (Figure 13). This is presumably due to the weaker basicity of tribenzylamine than that of *N*-Me-dibenzylamines. It has been reported that tribenzylamine is a weaker base than dibenzylamine and other aliphatic amines [78]. Such proton transfer is often seen in metal complexes of azacalixarenes [44,45,46,47,48].

The yellow color of DMF ⊂ **8** (X = *tert*-Bu, R = Me) seems to be the result of intramolecular charge transfer by phenolate formation. In each DMF inclusion, there is a clear C-H∙∙∙π interaction between the two hydrogens of the *N*-Me group of DMF and the two aromatic rings (Figure 13); the DMF molecule is encapsulated in the cavity from the bulkier *N*, *N*-dimethyl group rather than from the -CHO side. C-H∙∙∙π interaction is a type of weak hydrogen bonding between C-H and π electrons, which is also shown in some of the inclusion compounds of calix[n]arenes [79,80,81,82,83]. A search of the CSD crystal structure database shows that the C-H∙∙∙π non-bond distance averages 291 pm, and analysis of the protein database shows that the most frequent distance between the C and π planes is 370–380 pm [53,54,55,56,57,58,59,60,61]. The distance between the *N*-Me carbon and the two aromatic rings to the π-plane averages 346 pm in the three DMF ⊂ **8**. The distance between the *N*-Me hydrogen and the two aromatic rings to the π-plane averages 277 pm. In view of these facts, the C-H···π distance in DMF ⊂ **8** is clearly shorter than that of the database. Thus, for these azacalixarenes, the main interaction that envelops the DMF molecule is the C-H···π interaction.

### 5.2. Self-Inclusion Type Azacalixarenes by CH···π Interaction

It was shown that azacalix[3.1.1.1]arenes can encapsulate small molecules such as DMF and that CH···π interactions work between the host and guest. Thus, it was hypothesized that the introduction of an alkyl chain into the side chain would lead to the formation of a supramolecular structure connected by CH···π interactions in the crystal. Therefore, *p*-methyl-*N*-(4-phenylbutyl)azacalix[3.1.1.1]arene **8** (X = Me, R = (CH_2_)_4_Ph) with a phenylbutyl group as a side chain was synthesized in 20.5% yield (Figure 14).

Theoretical calculations indicate that the origin of this interaction is the dispersion force [84,85,86,87]. Although the CH···π interaction is a weak interaction, (4–14 kJ mol^−1^) [88,89,90,91,92,93,94], it can become non-negligible if many interactions are active at the same time [95,96,97]. Unlike interactions such as charge–charge and charge–dipole, the dispersion force is not affected by the polarity of the surrounding environment and can be observed both in solution and in solids. Therefore, it was inferred that the CH···π interaction is observed in the system of **8** (X = Me, R = (CH_2_)_4_Ph) both in solution and in the solid-state.

Since the side chains of the *N*-benzyl- and *N*-picolyl-azacalix[3.1.1.1]arene derivatives were located outside the ring structure, the author expected that the phenylbutyl group was also located outside the molecule (*exo* type). In fact, however, in the crystal structure, the phenylbutyl group was incorporated into the molecular cavity (*endo* type), and five of the eight hydrogens of the butyl group interacted with the aromatic ring forming the ring structure (CH···π interaction). In addition, one hydrogen of the benzene ring at the end of the phenylbutyl group was close to that of the π-plane of another molecule, where CH(Ar)···π interactions were also at work. In the crystal, a chain of CH(Ar)···π interactions was formed, and a CH(Ar)···π supramolecular structure was observed (Figure 15). In solution, the *exo* ⟷ *endo* equilibrium of the phenylbutyl group due to the CH···π interaction was found to be established by temperature-variable NMR from 25 to −60 °C. In addition, H-HCOSY was measured at −20 °C, and the signals of the *exo* and *endo* forms were assigned. This produced the ratio of the two isomers at each temperature and the Gibbs energy change for this interconversion, ∆*G* (273 K) = −3.1 kJmol^−1^ (∆*H* = −6.5 kJmol^−1^, ∆*S* = −35 Jmol^−1^). In solution, the endo-conformer was the minor component, and the *exo*-form was the major component. In the solid-state, it is the *endo*- conformer as the stabilization of the CH···π interaction overcomes the thermal energy.

## 6. Conclusions

Unlike other calixarenes, azacalixarenes contain nitrogen atoms in the cyclic skeleton, which makes it possible to introduce various side chains. The synthesis is simple, and the azacalix skeleton can be formed by several different methods. The yields are satisfactory. The presence of basic nitrogen and acidic phenolic hydroxyl groups in the cyclic skeleton itself gives it a unique metal ion selectivity. Compounds with relatively deep cavities, such as azacalix[3.1.1.1]arene, can include small molecules, such as DMF. The inclusion phenomenon must be studied in more depth. Another feature of azacalixarene is that it has high solubility. Even when small groups, such as halogen or methyl groups, are introduced into the para-position of phenol, azacalixarene is easily soluble in common solvents, making it easy to perform NMR and other measurements. It is also expected to be applied by converting not only the *N*-side chain but also the substituent of the benzene ring. Although it was not mentioned in this review, the author also succeeded in introducing azo groups into the *para*-position of phenol, which is expected to change the color by the complexation of metal ions. The synthesis of capsule molecules by the conversion of substituents on benzene rings is a future challenge. In addition, it was possible to observe various interactions using azacalixarenes, such as hydrogen bonding and CH∙∙∙π and π∙∙∙π interactions. The molecular design of azacalixarenes could be applied to the detection of interesting interactions, the assembly of supramolecular structures, and the highly selective extraction and complexation of metal ions.

## Data Availability

Not applicable.

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
