# Peer review of "Synthesis of Azacalixarenes and Development of Their Properties"

_molecules, 2021, doi:10.3390/molecules26164885_

Round 1

Reviewer 1 Report

The present paper highlights the synthesis, structure, interaction with metal ions, detection of some weak interactions using the structure, and construction of supramolecules of azacalixarenes. The review is interesting and the manuscript can be accepted for publication after addressing the following comments.

  1. Significance of this review is required?
  2. Some figures of molecules in Scheme 6, 8, 13, 14, 15 are too small.
  3. Some sentences are too long to understand. For example, Page 2, lines 39-42.
  4. The author should give the outlook.
  5. The perspective and future prospects of azacalixarenes should be discussed.
  6. Please check the grammar and spelling mistakes in the entire article.

Author Response

Reviewer #1

* Comments and Suggestions for Authors

  1. Significance of this review is required?
  2. Some figures of molecules in Scheme 6, 8, 13, 14, 15 are too small.
  3. Some sentences are too long to understand. For example, Page 2, lines 39-42.
  4. The author should give the outlook.
  5. The perspective and future prospects of azacalixarenes should be discussed.
  6. Please check the grammar and spelling mistakes in the entire article.

Ans.

  1. The previous reviews contain only old information. In this review, we have summarized new information, such as the discovery of a new synthetic method, the observation of intermolecular interactions by azacalixarene molecules, and their application to supramolecules. We were able to report the new properties and potential of azacalixarenes.
  2. I made the font as large as possible as the reviewer pointed out.
  3. I received proofreading from a native speaker.
  4. and 5. In this review, all work on azacalixarene has been completed. The author is planning to retire next year. Therefore, there are no further prospects.
  5. The paper has been proofread by a native speaker.

Reviewer 2 Report

The review article describes the synthesis and some of the properties of aza calixarenes. The author has been one of the major contributors to this highly specialized area and is therefore well positioned to contribute a review. I have a few comments for the author to consider, that I will list in what follows.

The first comment is that the English needs a lot of attention. Although I was able to piece together the meaning of what was written, this required effort.

In the discussion of Burke's 1949 paper on page 2, the author states that structure 1 "seems to be too strained for researchers studying cyclophanes". By writing "seems" it would appear that there is some doubt in the author's mind that a trans cyclohexene (compound 1 is a trans cyclohexene) is too strained to be stable. Obviously, structure 1 is wrong and this should be stated very clearly on page 2.

Also on page 2 and referring to a azacalixarene the author states that "NMR showed that it was a cyclic material" because of the broadness of the phenolic hydroxyl group that was hydrogen bonded. The compound could have been oligomeric and acyclic and it might still show the same feature. The hydrogen bonded OH alone cannot provide evidence that the compound is cyclic.

On page 6 the author states that "The tert-butyl group is too soluble and may cause problems." This is an example of where the English should be corrected. It is incorrect to state that the tert-butyl group is too soluble. The tert-butyl group confers enhanced solubility on the molecule.

On page 8 the statement is made that "scrambling of the phenol unit often occurs". This was puzzling to me, because the phenol unit isn't scrambled (as best I can determine), but one phenol unit can be cleaved from the oligomeric fragment (see Scheme 5). Some clearer explanation will be appreciated.

On page 10 on the line right after Scheme 7, the author should point the reader to Scheme 8 so that the reader knows where to look to find the structures that are being discussed. This issue arises in several places: the number of the Scheme or Figure should be mentioned at the first sentence that refers to a compound in the Scheme or Figure.

In Scheme 8 the bottom most equation must be corrected to show tris(3-aminoethyl)amine as the starting material that leads to 25a. (Note that in the title of Scheme 8 "Synthesis" is misspelled.)

I was puzzled by the mechanism the author proposes in Scheme 10. The process shown is concerted and must take place through an 8-membered ring transition state. This seems less likely than an alternative mechanism involving reversible proton transfer from the phenol to the oxygen atom of the oxazine, cleavage to the iminium ion that undergoes nucleophilic attack by the activated phenoxide. Note also that there is an error in the reference 18: the correct journal is JOC and not JACS.

On page 16, Scheme 16, the structure of 32 should be corrected so the two OH groups do not overlap.

On page 17, Table 2, free energy data are included for ten compounds, presumably associated with a conformational change of the type shown in Figure 5 on page 18 for one of the compounds. Each of the compounds listed in Table 2 presumably has many energetically accessible conformations so it is not clear what the numbers refer to. Also, there is no discussion in the text of any of these numbers nor of their significance. The author should either add some discussion so the reader can understand what the significance is of the free energy changes, or the author should delete that data from Table 2.

Following the discussion of the solution phase structures of the azacalixarenes, the author discusses solid phase structures of some inclusion complexes. It would be helpful to include some discussion to let the reader know whether there is any relationship between solution phase and solid phase structures.

I recommend that this manuscript be published after the author addresses the issues mentioned above.

Author Response

#1 The first comment is that the English needs a lot of attention. Although I was able to piece together the meaning of what was written, this required effort.

Ans. The paper has been proofread by native speaker.

#2 In the discussion of Burke's 1949 paper on page 2, the author states that structure 1 "seems to be too strained for researchers studying cyclophanes". By writing "seems" it would appear that there is some doubt in the author's mind that a trans cyclohexene (compound 1 is a trans cyclohexene) is too strained to be stable. Obviously, structure 1 is wrong and this should be stated very clearly on page 2.

Ans. Burke et al. showed the structure of aza[3]metacyclophane 1 (Figure 1), but their experiments proved this to be incorrect. I have included the text on this on the second page, along with a response to the following comment. Incidentally, [3]metacyclophane and [4]metacyclophane have not been synthesized to date.

#3 Also on page 2 and referring to a azacalixarene the author states that "NMR showed that it was a cyclic material" because of the broadness of the phenolic hydroxyl group that was hydrogen bonded. The compound could have been oligomeric and acyclic and it might still show the same feature. The hydrogen bonded OH alone cannot provide evidence that the compound is cyclic.

Ans. It seemed clear to the author that the OH signal was not as broad as this, even if it was an oligomer, but the author changed the text to conform to the reviewer's wishes. If it were acyclic, there would clearly be a terminal signal, but it was not found. In addition, the symmetry of the spectrum showed that the structure was definitely cyclic. I added this sentence to p2 as follows:

This structure, [3]metacyclophane, seems to be clearly distorted to cyclophane researchers. Therefore, it seems obvious that it cannot be produced by their synthetic method. In fact, Burke et al. experimentally ruled out the structure of 1. Furthermore, it is clear to the author that the OH signal is not as broad as it should be, even for oligomers. Moreover, if it were acyclic, a terminal signal would be shown, but this was not found. The symmetry of the spectra also confirmed that the structure was cyclic. Then, the author estimated that if it were to form, it would be a trimer.

#4 On page 6 the author states that "The tert-butyl group is too soluble and may cause problems." This is an example of where the English should be corrected. It is incorrect to state that the tert-butyl group is too soluble. The tert-butyl group confers enhanced solubility on the molecule.

Ans. The reviewer makes a good point. This was rewritten as follows:

The introduction of tert-butyl group makes the azacalixarens too soluble and may cause problems.

#5 On page 8 the statement is made that "scrambling of the phenol unit often occurs". This was puzzling to me, because the phenol unit isn't scrambled (as best I can determine), but one phenol unit can be cleaved from the oligomeric fragment (see Scheme 5). Some clearer explanation will be appreciated.

Ans. Scrambling has been removed and replaced with “Cleavage”.

#6 On page 10 on the line right after Scheme 7, the author should point the reader to Scheme 8 so that the reader knows where to look to find the structures that are being discussed. This issue arises in several places: the number of the Scheme or Figure should be mentioned at the first sentence that refers to a compound in the Scheme or Figure.

Ans. I have corrected it as the reviewer pointed out (P 11, L 213). However, I do not expect the same to be true elsewhere, as I have tried to write in a way that is clear to the reader.

#7 In Scheme 8 the bottom most equation must be corrected to show tris(3-aminoethyl)amine as the starting material that leads to 25a. (Note that in the title of Scheme 8 "Synthesis" is misspelled.)

Ans. Corrections were made.

# 8 I was puzzled by the mechanism the author proposes in Scheme 10. The process shown is concerted and must take place through an 8-membered ring transition state. This seems less likely than an alternative mechanism involving reversible proton transfer from the phenol to the oxygen atom of the oxazine, cleavage to the iminium ion that undergoes nucleophilic attack by the activated phenoxide. Note also that there is an error in the reference 18: the correct journal is JOC and not JACS.

Ans. The author's mechanism was removed, and Burke's mechanism was inserted instead. Journal corrections were made.

#9 On page 16, Scheme 16, the structure of 32 should be corrected so the two OH groups do not overlap.

Ans. Corrections were made as pointed out by the reviewer.

#10 On page 17, Table 2, free energy data are included for ten compounds, presumably associated with a conformational change in the type shown in Figure 5 on page 18 for one of the compounds. Each of the compounds listed in Table 2 presumably has many energetically accessible conformations so it is not clear what the numbers refer to. Also, there is no discussion in the text of any of these numbers nor of their significance. The author should either add some discussion so the reader can understand what the significance is of the free energy changes, or the author should delete that data from Table 2.

Ans. It is clear to researchers working with calixarene that this energy difference is due to the cone«partial cone. The molecule is rigid when these values are large. The reason for this is hydrogen bonding between hydroxyl groups, since the structure of the macrocyclic skeleton is the same. On p19, it is mentioned that "The strength of this hydrogen bond also affects the conformational change in the molecule. In other words, the barrier of aromatic ring inversion (cone«partial cone) is higher for azacalix[3.1.1.1]arene than for calix[4]arenes (Table 2)".

#11 Following the discussion of the solution phase structures of the azacalixarenes, the author discusses solid phase structures of some inclusion complexes. It would be helpful to include some discussion to let the reader know whether there is any relationship between solution phase and solid phase structures.

Ans The reviewer has a good point. I am unsure how to make such an argument. How can we determine the relationship between the structure in the solid phase and the structure in the liquid phase? If it is possible, please let me know.

I recommend that this manuscript be published after the author addresses the issues mentioned above.

Round 2

Reviewer 2 Report

In the corrected manuscript most of the suggestions for change that I made have been addressed.

Please correct reference 18. The correct journal is JOC and not JACS. [From first review --> Note also that there is an error in the reference 18: the correct journal is JOC and not JACS.]

On line 146 the M should be capitalized in Ome.

Burke's mechanism, the one shown in Scheme 10 has several problems and is unlikely to convince anyone. Neither does showing this add anything constructive to the review. My suggestion would be to remove this.

Author Response

In the corrected manuscript most of the suggestions for change that I made have been addressed.

Please correct reference 18. The correct journal is JOC and not JACS. [From first review --> Note also that there is an error in the reference 18: the correct journal is JOC and not JACS.]

Ans: Corrected. JACS -> JOC

On line 146 the M should be capitalized in Ome.

Ans. Corrected.

Burke's mechanism, the one shown in Scheme 10 has several problems and is unlikely to convince anyone. Neither does showing this add anything constructive to the review. My suggestion would be to remove this.

Ans: Scheme 10 was deleted.